# Biweekly CAPOX versus triweekly CAPOX in the adjuvant therapy of post-surgery CRC: A randomized controlled trial

**Hangyu Zhang**[1]�, **Danyang Wang**[2]☻, **Zhou Tong**[1], **Tao Xiang**[2], **Xudong Zhu**[1], **Lulu Liu**[1], **Yi Zheng**[1], **Peng Zhao**[1], **Weijia Fang**[1], **Wenbin Chen**[2]*

**1** Department of Medical Oncology, First Affiliated Hospital, School of Medicine, Zhejiang University, Hangzhou, Zhejiang, P. R. China, **2** Department of colorectal surgery, First Affiliated Hospital, School of Medicine, Zhejiang University, Hangzhou, Zhejiang, P. R. China

☻ These authors contributed equally to this work.
* wenbinchen@zju.edu.cn

## Abstract

### Purpose

This study aims to compare the safety and efficiency of modified biweekly CAPOX and conventional triweekly CAPOX in high-risk stage II and stage III post-surgery colorectal (CRC) patients.

### Methods

From July 25, 2018, to May 14, 2021, high-risk stage II and stage III post-surgery CRC patients were randomized in the control triweekly group (intravenous infusion of oxaliplatin 130 mg/m$^2$ on day 1 and oral capecitabine 1000 mg/m$^2$, twice daily from day 1 to day 14) and the experimental biweekly group (intravenous infusion of oxaliplatin 85 mg/m$^2$ on day 1 and oral capecitabine 1000 mg/m$^2$, twice daily from day 1 to day 10). The primary endpoint was the incidence rate of thrombocytopenia. The secondary endpoint was 3-year disease free survival (DFS) rate. The patients follow up was started on July 25, 2018, and finished on October 8, 2024.

### Results

A total of 160 patients were 1:1 randomly assigned (80 patients to biweekly group and 80 patients to triweekly group). All grade thrombocytopenia occurred in 33% and 49% patients at biweekly and triweekly group, respectively ($P = 0.02$). Neutropenia presented in 36% and 51% patients at biweekly and triweekly group, respectively ($P = 0.04$). The second endpoint 3-year DFS was 85.1% in biweekly group and 80.4% in triweekly CAPOX group ($P = 0.51$, HR = 0.78, [95%CI, 0.38–1.63]). The total rate of uncomplete therapy patient was 7.5% and 15% in biweekly and triweekly group, respectively ($P = 0.13$).

**Data availability statement:** All relevant data are within the paper and its Supporting Information files.

**Funding:** Zhejiang Provincial National Science Foundation of China LQ23H160041 (to H.Z).

**Competing interests:** The authors have declared that no competing interests exist.

## Conclusion

Biweekly CAPOX presented significant less thrombocytopenia and neutropenia than triweekly CAPOX regimen. There was no difference in 3-year DFS between biweekly CAPOX and triweekly CAPOX.

**Clinical trial registration:** ClinicalTials.gov (NCT03564912).

## Introduction

Colorectal cancer (CRC) was estimated to be the fourth most common cancer and the second leading cause of cancer deaths worldwide [1]. A large proportion of CRC patients are diagnosed in advanced stage cause no warning symptoms [2]. Surgery remains the foundation of curative treatment, and perioperative chemotherapy, especially postoperative adjuvant chemotherapy plays particularly important role. To improve overall and disease-free survival, adjuvant chemotherapy is widely applied in high-risk stage II and stage III CRC patients.

Since the MOSAIC study [3], CAPOX (capecitabine and oxaliplatin) and FOLFOX (fluorouracil plus leucovorin and oxaliplatin) had become the standard adjuvant regimens for high-risk stage II and stage III CRC [4]. However, whether adjuvant chemotherapy improves clinical outcomes of patients with stage II or curatively resected stage IV colon cancer, and whether duration lasts for 3 months or 6 months, are still controversial. Recently years, a few studies [5,6] have focused on the adjuvant treatment duration course, hoping to reduce treatment-related adverse events and provide intact clinical benefits. Peripheral sensory neuropathy (PSN) is the most criticized adverse event in these studies and its incidence was significantly lower with 3-month therapy than with 6-month therapy [7,8]. Indeed, hematologic toxicity, especially thrombocytopenia, is an equally important factor affecting the integrity of adjuvant treatment. It is reported that grade ≥3 thrombocytopenia was about 5% in CAPOX adjuvant treated patients [9], compared with 0–1.7% in FOLFOX adjuvant treated patients [3,5]. Previous studies have not paid sufficient attention to the difference between the two treatment regimens and the proportion of thrombocytopenia leading to treatment intolerance. Indeed, biweekly CAPOX was reported to have no incidence of grade 3 thrombocytopenia in first-line treatment of metastases CRC in several studies [10,11]. This may be related to a modified single dose of oxaliplatin, which has not been validated in adjuvant therapy.

Therefore, we conducted this pilot study of biweekly CAPOX versus triweekly CAPOX for high-risk stage II and stage III colon cancer to assess the potential implications in modified single dose oxaliplatin for Asian patients. One of the key objectives of our study was to provide data to confirm the treatment related adverse events difference between the two regimens and its impact on treatment tolerability. Another objective was to explore whether modified biweekly CAPOX could affect disease free survival (DFS) in adjuvant CRC patients.

## Materials and methods

### Study design

This study is an open-label, randomized, single-center pilot study conducted in China. Patients who had complete resection for high-risk stage II and stage III colon cancer were randomized 1:1 assigned to receive either biweekly CAPOX or triweekly CAPOX. The duration time of chemotherapy refer to the recently published adjuvant studies [12] (high-risk stage II and low-risk stage III population received 3-month CAPOX therapy, high-risk stage III population received 6-month CAPOX therapy). The definition of high-risk stage II colorectal cancer includes the following factors: 1. Number of lymph nodes analyzed: If fewer than 12 lymph nodes were analyzed post-surgery. 2. Poor prognostic features: Poorly differentiated histology (exclusive of those that are MSI-H); Lymphatic/vascular invasion; Bowel obstruction; Perineural invasion (PNI); Localized perforation; Close, indeterminate, or positive surgical margins.

The inclusion criteria of patients were curatively resection of high-risk stage II and stage III colon cancer patients, the age range from 18–75 years, ECOG performance status 0–2, adequate renal, hepatic and bone marrow function. The major exclusion criteria were those who received neoadjuvant treatment before surgery or unstable heart disease, active inflammation, etc. The Ethics Committee of the First Affiliated Hospital of Zhejiang University School of Medicine approved the protocol. The need for consent was waived by the ethics committee.

The authors acknowledge that the trial was registered after participant recruitment started due to logistical delays. We confirm that all ongoing and related trials for this intervention are registered (NCT 03564912). Participant recruitment started on July 25, 2018, and finished on May 14, 2021. The follow up was started on July 25, 2018, and finished on October 8, 2024.

The primary endpoint was incidence of thrombocytopenia. The secondary endpoint was 18-month DFS rate defined as the proportion of patients who relapse or death from any case patients within 18 months.

### Treatment

Patients were enrolled by key sub-I H.Y.Z, then 1:1 randomized to the biweekly group or triweekly group via blocked randomization with random block size of 4 to balance enrolled participants by PI (W.J.F). Triweekly CAPOX include intravenous infusion of oxaliplatin 130 mg/m$^2$ on day 1 and oral capecitabine 1000 mg/m$^2$, twice daily from day 1 to day 14, biweekly CAPOX include intravenous infusion of oxaliplatin 85 mg/m$^2$ on day 1 and oral capecitabine 1000 mg/m$^2$, twice daily from day 1 to day 10. Patients were evaluated for disease recurrence by abdominal computed tomography (CT) scans and serum CEA every 3 months, chest CT scans every 6 months, and colonoscopy every 12 months. Adverse events were monitored from the first day of treatment initiation until 28 days after the last dose of CAPOX. All events were assessed according to the Common Terminology Criteria for Adverse Events Version (CTCAE) version 4.0, and the incidence of all-grade thrombocytopenia during this observation window was recorded as the primary safety endpoint. Therapeutic dose was planned to reduce by 20% when grade ≥ 3 adverse events happened.

### Sample size

Sample size calculation was performed with thrombocytopenia as the primary outcome, use a significant level of 5% and a power level of 80%. 80 participants were required for each group based on 0.44 effect size.

### Statistical analysis

DFS curve was derived by Kaplan-Meier estimation. Cox proportional hazard model was used to calculate the HRs and 95% CIs between biweekly group and triweekly group. The AE frequency was compared using chi-square test. For all tests, P < 0.05 was defined as statistically significant. The IBM SPSS Statistics (Version 26; IBM Corp., New York, USA) was used for the analyses. The GraphPad Prism 8 (GraphPad Software, Inc., La Jolla, CA, USA) was used for chart making.

## Results

### Patients

Between July 25, 2018, and May 14, 2021, 160 patients were enrolled in our study 1:1 randomized received biweekly CAPOX or triweekly CAPOX adjuvant chemotherapy as shown in CONSORT diagram (Fig 1). All enrolled patients received at least one cycle therapy and were included in final analysis. The last follow-up time was October 8, 2024, and the median follow-up period was 42 months.

The baseline characteristics were shown in Table 1. There were 80 patients in biweekly group and another 80 patients in triweekly group, with balanced characteristics between two groups. Sixty- three percent of patients (100/160) were high-risk stage II and low-risk stage III patients who received 3-month treatment, and thirty-seven percent of patients were high-risk stage III patients who received 6-month treatment. There were 47 patients had right-side primary tumor and 103 patients had left-side primary tumor. The proportion of patients who had elevated serum CEA and CA19−9 before surgery were 30% and 19%, respectively.

### Thrombocytopenia and other adverse events

All grade thrombocytopenia occurred 33% in biweekly group and 49% in triweekly group ($P=0.02$). In the biweekly group, the incidence of thrombocytopenia was 28% in the 3-month treatment subgroup and 38% in the 6-month treatment sub-group. In the triweekly group, the incidence of thrombocytopenia was 43% in the 3-month treatment subgroup and 58% in the 6-month treatment subgroup. However, thrombocytopenia more than grade 3 presented 5% versus. 9% in biweekly and triweekly group with no significant difference ($P=0.35$).

The adverse events summary is shown in the Table 2 and Fig 2. The triweekly CAPOX treatment group demonstrated significantly higher adverse events rate than biweekly CAPOX treatment group, especially in all grade neutropenia ($P=0.04$). However, severe neutropenia (grade ≥ 3) was similar in triweekly and biweekly group ($P=0.17$). In patients receiving 6-month CAPOX, 21% (13/60) of the patients experienced grade ≥ 2 PSN, whereas 10% (10/100) of the patients in the 3-month CAPOX group experienced grade ≥2 PSN. There was no treatment related death.

We also evaluated the treatment completion and management of adverse events. The biweekly group showed a relatively low rate of treatment interruption ($P=0.13$). In the biweekly group, 6 patients (7.5%) failed to complete adjuvant chemotherapy due to treatment intolerance. Whereas, in the triweekly group, the adjuvant treatment discontinue patients was 12 (15%), including 11 patients with treatment intolerance and 1 patient found to have leukemia during the course. Additionally, the proportion of patients requiring dose reductions was 13% in the biweekly regimen group and 28% in

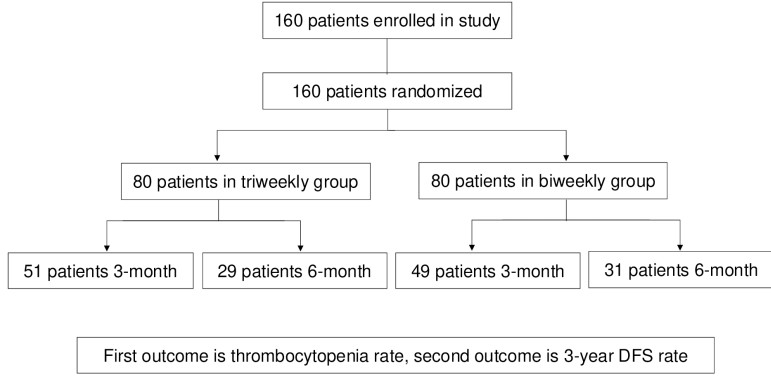

**Fig 1. CONSORT flow diagram.**

**Table 1. Characteristics of all enrolled patients.**

| | Biweekly CAPOX (n=80) | Triweekly CAPOX (n=80) | P value |
|---|---|---|---|
| Duration of treatment | | | 0.74 |
| 6-month | 31 (38) | 29 (36) | |
| 3-month | 49 (62) | 51 (64) | |
| Age, median(range), years | 58 (32-77) | 57 (24-78) | 0.48 |
| <70 | 68 (85) | 71 (88) | |
| ≥70 | 12 (15) | 9 (12) | |
| Sex | | | 0.19 |
| Male | 45 (56) | 53 (66) | |
| Female | 35 (46) | 27 (34) | |
| Location of primary tumor | | | 0.19 |
| Right | 25 (31) | 22 (27) | |
| Left | 52 (65) | 51 (64) | |
| NA | 3 (4) | 7 (9) | |
| T stage | | | 0.16 |
| T1-3 | 68 (85) | 61 (76) | |
| T4 | 12 (15) | 19 (24) | |
| N statge | | | 0.55 |
| N0 | 16 (20) | 17 (21) | |
| N1 | 40 (50) | 45 (56) | |
| N2 | 24 (30) | 18 (23) | |
| Number of haversted lymph nodes | | | 0.32 |
| <12 | 2 (2) | 0 (0) | |
| ≥12 | 70 (88) | 72 (90) | |
| NA | 8 (10) | 8 (10) | |
| TNM stage | | | > 0.99 |
| Stage II | 20 (25) | 19 (24) | |
| Stage IIIA | 36 (45) | 35 (44) | |
| Stage III B | 15 (19) | 10 (12) | |
| Stage III C | 9 (11) | 16 (20) | |
| CEA before surgery | | | 0.79 |
| <5 | 45 (56) | 47 (59) | |
| ≥5 | 26 (32) | 22 (57) | |
| NA | 9 (12) | 11 (88) | |
| CA199 before surgery | | | 0.83 |
| <37 | 56 (70) | 54 (68) | |
| ≥37 | 15 (19) | 15 (19) | |
| NA | 9 (11) | 11 (13) | |
| Recurrence patient | 13 (16) | 9 (11) | 0.35 |
| Uncomplete therapy patient | 6 (7) | 12 (15) | 0.13 |

the triweekly regimen group. The proportion of patients with treatment delays was 10% in the biweekly regimen group and 26% in the triweekly regimen group, with the average treatment delay being 3 days and 5 days, respectively. These data highlight the differences in patient tolerance to the two treatment regimens and the challenges in managing adverse events.

**Table 2. Adverse events by regimen and treatment duration in all patients.**

| Adverse events | Biweekly CAPOX group (n = 80) | | | Triweekly CAPOX group (n = 80) | | | P value |
|---|---|---|---|---|---|---|---|
| | 3-month treatment (N = 49) | 6-month treatment (N = 31) | Total | 3-month treatment (N = 51) | 6-month treatment (N = 29) | Total | |
| Hematological | | | | | | | |
| Neutropenia | 19(38) | 10(32) | 29(36) | 24(47) | 17(58) | 41(51) | 0.040 |
| Neutropenia (≥3) | 1(2) | 0 | 1(1) | 2(4) | 2(7) | 4(5) | 0.173 |
| Thrombocytopenia | 14(28) | 12(38) | 26(33) | 22(43) | 17(58) | 39(49) | 0.027 |
| Thrombocytopenia (≥3) | 1(2) | 3(10) | 4(5) | 5(10) | 2(7) | 7(9) | 0.349 |
| Anemia | 10(20) | 8(25) | 18(22) | 7(14) | 13(45) | 20(25) | 0.710 |
| Non-hematological | | | | | | | |
| PNS | 24(49) | 16(52) | 40(50) | 32(62) | 15(52) | 47(58) | 0.267 |
| PNS (≥2) | 5(10) | 6(19) | 11(14) | 7(14) | 4(14) | 11(14) | 1.0 |
| Liver disfunction | 15(30) | 7(22) | 22(27) | 22(43) | 10(34) | 32(40) | 0.095 |
| Hyperbilirubinemia | 5(10) | 6(19) | 11(14) | 5(10) | 7(24) | 12(15) | 0.822 |
| Diarrhea (≥3) | 0 | 1(3) | 1(1) | 0 | 1(3) | 1(1) | 1.0 |
| Vomiting (≥3) | 0 | 0 | 0 | 2(4) | 0 | 2(2) | 0.155 |

## Disease-free survival

At the time of analysis, 29 events had been reported (13 in the biweekly group and 16 in the triweekly group). The Kaplan-Meier plots for DFS among biweekly group and triweekly group and subgroups are shown in Fig 3. The 3-year DFS rate was 85.1% [95% CI: 76.9%–93.3%] in biweekly group and 80.4% [95% CI: 71.2%–89.7%] in triweekly group ($P = 0.51$, HR = 0.78, [95%CI, 0.38–1.63]).

In the subgroup analysis of patients of 3-month treatment duration, the 3-year DFS was 89.2% [95% CI: 80.2%–98.2%] in biweekly group and 89.0% [95% CI: 79.8%–98.1%] in triweekly group (Fig 4A). The difference was not statistically significant (P = 0.643). The HR for biweekly group to triweekly group was 0.89 [95% CI: 0.29–2.77], indicating no significant difference in DFS between the two groups.

For patients of 6-month treatment duration, the 3-year DFS was 77.6% [95% CI: 61.6%–93.6%] in biweekly group and 67.7% [95% CI: 50.3%–85.1%] in triweekly group (Fig 4B), the HR was 0.79 [95% CI: 0.30–2.10] with no significant difference observed between the two groups (P = 0.800).

The median disease-free survival time of 29 relapse patients is 21 months (range 2.7–53.7 months). Up to the last follow-up, 3 patients in the triweekly group died due to disease progression, while no deaths were reported in the biweekly group.

## Discussion

Our study is the first study explore the safety and efficacy of biweekly CAPOX and triweekly CAPOX in colorectal adjuvant treatment. Previous study had shown that biweekly CAPOX had modest efficacy and an acceptable safety profile in the treatment of advanced gastric cancer [13]. In our study, we enrolled high-risk stage II and stage III post-surgery colorectal cancer patients to certify its advantage of biweekly CAPOX in adjuvant area.

CAPOX is reported to have significant higher grade 3–4 thrombocytopenia than FOLFOX (3–12% vs. 1–6%) [14]. However, the negative effect of thrombocytopenia induced by CAPOX has not been paid enough attention. Indeed, not only grade 3–4 thrombocytopenia affects treatment, but grade 2 thrombocytopenia also requires drug intervention and may lead to delayed treatment [15]. Our study finds that all grade thrombocytopenia significant reduced by modified

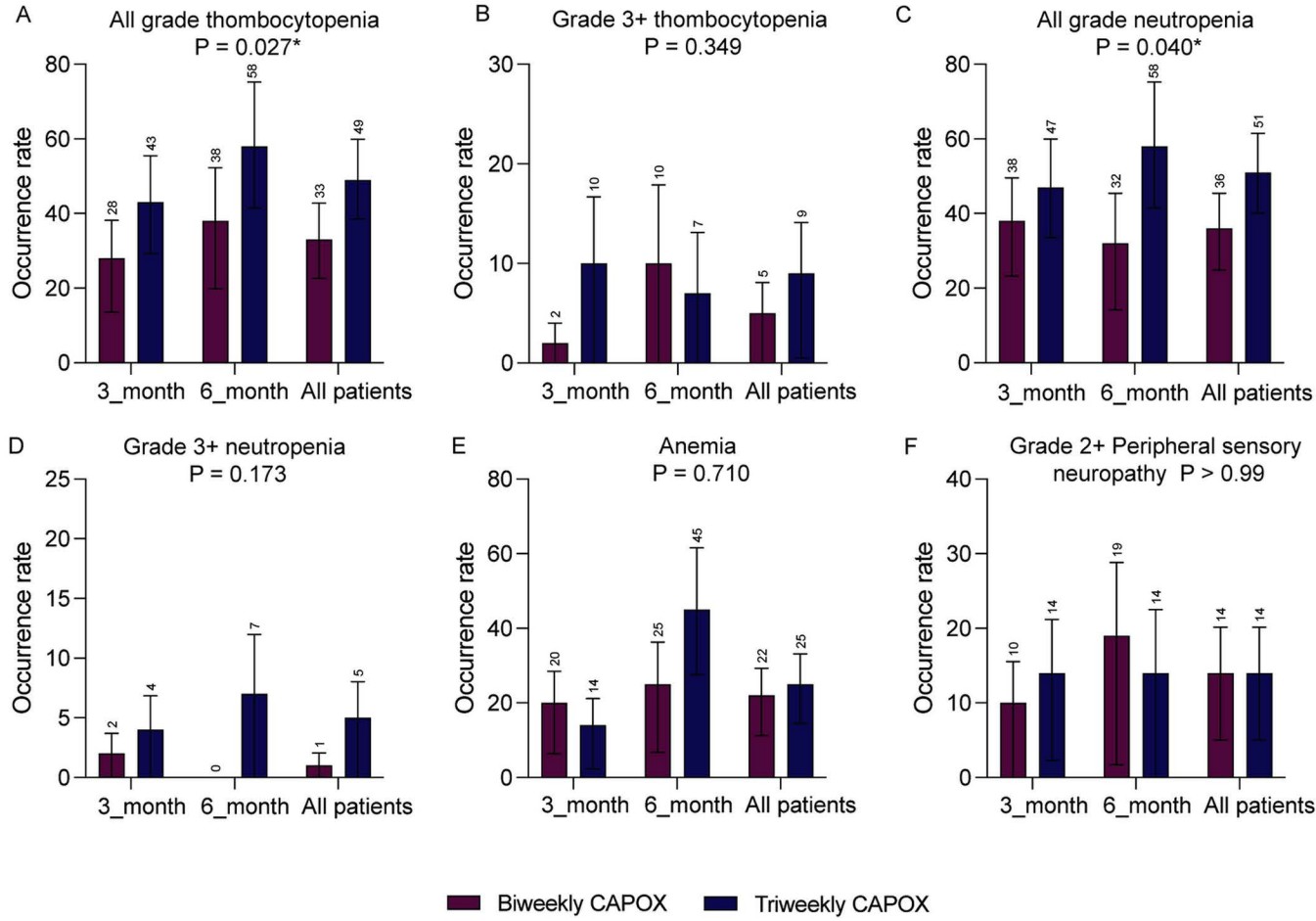

**Fig 2. Adverse events in biweekly and triweekly groups. (A)** All grade thrombocytopenia in 3-month, 6-month and all patients between biweekly and triweekly group. **(B)** Grade more than 3 thrombocytopenia rate between biweekly and triweekly group. **(C-D)** All grade neutropenia and grade more than 3 neutropenia rate in 3-month, 6-month and all patients between biweekly and triweekly group. **(E)** Anemia rate in 3-month, 6-month and all patients between biweekly and triweekly group. **(F)** Peripheral sensory neuropathy rate in 3-month, 6-month and all patients between biweekly and triweekly group.

biweekly CAPOX, which has not been explored before. The incidence of all hematological AEs was lower in the biweekly group than in the triweekly group, concomitant lower treatment discontinuation. In the biweekly CAPOX group, 74 patients (92.5%) completed all courses of adjuvant chemotherapy, which is higher than triweekly group and previous adjuvant study [16]. The completion rate of treatment is considered to be related to OS benefits [16]. The high completion rate of our study due to the high proportion of patients who underwent three-month treatment and the reduction of AEs of the modified biweekly CAPOX regimen. And it is shown that grade ≥ 2 PSN was more common in 6-month treatment patients, which was consistent with the finding of previous studies [17,18]. Given the limited sample size, the observed differences in toxicity should be considered exploratory. Although not practice-changing, these findings may provide a rationale for future studies evaluating biweekly oxaliplatin regimens in patients with poor treatment tolerance.

The 3-year DFS of CRC has been reported around 80% [19,20], which is consistent with our results. The DFS curves of biweekly and triweekly groups were similar in our study, and the 3-year DFS rate did not reach significant difference between the two groups (85.1% vs. 80.4%). It is reported that the median DFS in relapsed patients aged less than 50

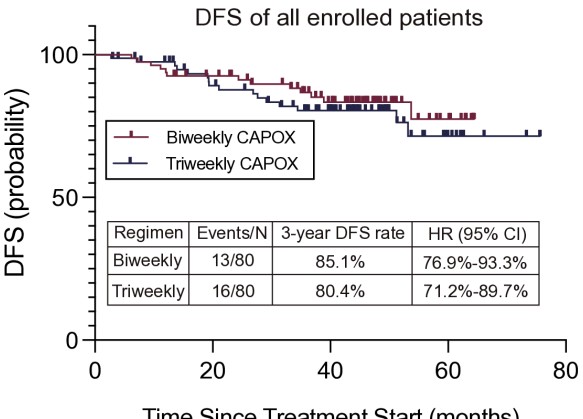

**Fig 3. Kaplan-Meier survival curves comparing Disease-Free Survival (DFS) for Biweekly and Triweekly CAPOX regimens.** The 3-year DFS rates were 85.1% for Biweekly and 80.4% for Triweekly CAPOX.

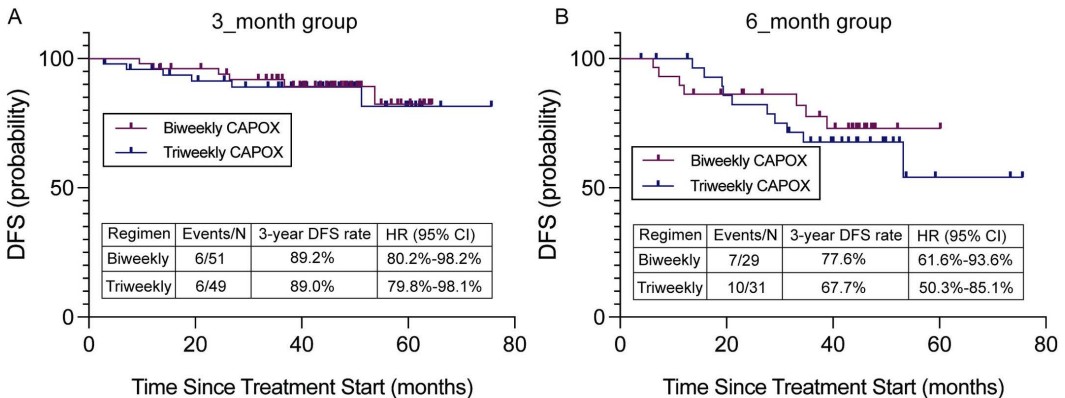

**Fig 4. Kaplan-Meier survival curves comparing Disease-Free Survival (DFS) for Biweekly and Triweekly CAPOX regimens in two subgroups: (A) The 3-year DFS rates for the 3-month group were 89.2% for Biweekly and 89.0% for Triweekly CAPOX. (B)** In the 6-month group, the 3-year DFS rates were 77.6% for Biweekly and 67.7% for Triweekly CAPOX.

years was 13 months [21]. And in another retrospective study, the median recurrence free survival was 1.3 years [22]. In our study, the median DFS was 21 months in relapse patients, which may be related to the high completion rate of adjuvant therapy. However, the overall number of patients was too limited to establish whether biweekly CAPOX is inferior to triweekly CAPOX.

In this study, we observed that the biweekly regimen was associated with a reduced incidence of hematological toxicities, including thrombocytopenia, compared to the triweekly regimen. While we did not directly assess hospitalization rates or healthcare costs, it is plausible that the reduction in toxicity may lead to improved patient quality of life by minimizing severe side effects, allowing for better physical function and fewer disruptions to daily activities. Additionally, the lower toxicity in the biweekly group might theoretically contribute to higher treatment adherence, as fewer patients required treatment discontinuation, dose reductions, or experienced treatment delays. Based on these observations, it is reasonable to speculate that the biweekly regimen could also offer a more cost-effective option by potentially reducing healthcare resources needed for managing severe toxicities. These findings suggest that the biweekly regimen could be a preferable

treatment option for certain patient populations, particularly those at higher risk for adverse events. Ultimately, these results provide important insights that may influence clinical guidelines, offering a well-tolerated and effective alternative for patients who need continuous therapy.

There may have new revolution of adjuvant chemotherapy of CRC in the future. More and more studies support the use of ctDNA test to identify patients who are at increased risk of recurrence and are potential benefit from treatment [23,24]. And it is reported that tumor deposits (TD) positive patients proven to have significantly worse outcomes, especially in N1a-b patients [25]. The role of TD deserves further scrutiny. And the best adjuvant treatment option of MSI-H CRC is still inconclusive. Although post-surgery CRC patients should be segmented in different groups, adjuvant treatment is still on the base of fluorouracil and oxaliplatin. We believe that modified biweekly CAPOX can reduce treatment-related AEs and turn out to be a better tolerance regimen for most CRC patients.

However, as a pilot study with a limited number of patients, it lacks the statistical power to definitively demonstrate the superiority of biweekly CAPOX over triweekly CAPOX. Further studies are warranted to confirm the potential advantages of the biweekly regimen. Additionally, this study is limited by its single-center design, which may introduce potential selection bias and limit the generalizability of the findings to other populations or healthcare settings. The sample size and setting may not fully represent the broader patient population, and the results may not be applicable to patients with different demographic characteristics or in different clinical contexts. As such, while our findings provide important insights, larger, multi-center studies are needed to validate these results and to assess the broader applicability of the biweekly regimen across diverse patient groups.

## Conclusion

Our pilot study suggests that the biweekly CAPOX regimen may be a viable alternative to the triweekly regimen, especially for patients with oxaliplatin intolerance. However, these findings need to be confirmed in larger, multicenter trials. Future research could also explore the use of biomarkers or molecular profiling to personalize adjuvant chemotherapy regimens and optimize treatment outcomes.

## Supporting information

**S1 File.  Clinical and survival data of all enrolled patients.**
(XLSX)

**S2 File.  English version of the study protocol.**
(PDF)

**S3 File.  Chinese version of the study protocol.**
(PDF)

## Acknowledgments

This study would like to appreciate Ms. Shiting Wen for her support and every participant enrolled in this study.

## Author contributions

**Conceptualization:** Hangyu Zhang, Danyang Wang.

**Data curation:** Zhou Tong, Tao Xiang, Xudong Zhu, Lulu Liu.

**Formal analysis:** Zhou Tong, Tao Xiang, Xudong Zhu, Lulu Liu.

**Supervision:** Yi Zheng, Peng Zhao, Weijia Fang.

**Writing – original draft:** Hangyu Zhang.

**Writing – review & editing:** Wenbin Chen.

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
