## [Decision Letter · Decision Letter 0]

PONE-D-24-47321Biweekly CAPOX versus Triweekly CAPOX in the adjuvant therapy of post-surgery CRC: a randomized controlled trialPLOS ONE

Dear Dr. Zhang,

Thank you for submitting your manuscript to PLOS ONE. After careful consideration, we feel that it has merit but does not fully meet PLOS ONE’s publication criteria as it currently stands. Therefore, we invite you to submit a revised version of the manuscript that addresses the points raised during the review process.

We look forward to receiving your revised manuscript.

Kind regards,

Keun-Yeong Jeong

Academic Editor

PLOS ONE

Journal Requirements:

2. Thank you for submitting your clinical trial to PLOS ONE and for providing the name of the registry and the registration number. The information in the registry entry suggests that your trial was registered after patient recruitment began. PLOS ONE strongly encourages authors to register all trials before recruiting the first participant in a study.

a) your reasons for your delay in registering this study (after enrolment of participants started);

b) confirmation that all related trials are registered by stating: “The authors confirm that all ongoing and related trials for this drug/intervention are registered.

“This work was supported by Natural Science Foundation of Zhejiang Province, No. LQ23H160041.”

“Zhejiang Provincial National Science Foundation of China LQ23H160041 (to H.Z)”

Reviewers' comments:

Reviewer's Responses to Questions

**Comments to the Author**

1. Is the manuscript technically sound, and do the data support the conclusions?

Reviewer #1: Yes

Reviewer #2: Yes

Reviewer #3: Yes

2. Has the statistical analysis been performed appropriately and rigorously? 

Reviewer #1: Yes

Reviewer #2: Yes

Reviewer #3: Yes

3. Have the authors made all data underlying the findings in their manuscript fully available?

Reviewer #1: Yes

Reviewer #2: Yes

Reviewer #3: Yes

4. Is the manuscript presented in an intelligible fashion and written in standard English?

Reviewer #1: Yes

Reviewer #2: Yes

Reviewer #3: No

5. Review Comments to the Author

Reviewer #1: Thank you for an interesting article that clearly outlines what was done and how to interpret the findings. I have a few comments to further clarify the article:

Abstract: "This study aims" - please add an s.

line 139: 0.44 effect size - isn't this a calculation based on a difference in proportions between the two groups? What proportions were used? was the proportion of all grade or grade 3+ the quantity of interest? Please provide a calculation that matches the incidence endpoint. Be clear if it is a rate estimated at any time during treatment, or if it is calculated from a time to event variable, which is possible if the AEs can take time to be experienced and there is differing times of follow up.

Non-inferiority is mentioned in the discussion - if the study was designed to test non-inferiority, please clearly indicate this in the sample size section of the paper.

What happened if a patient in the 3-month group experienced thrombocytopenia at 4 months - was that counted as an event? Be clear if the adverse events were collected within 6 months of starting treatment or just during study treatment.

Figure 2F - state p>0.99

The description of the results is valid.

Could 95% confidence intervals be added to the rates in Figures 2 and 3?

Reviewer #2: The authors present their single center randomized phase II study for high risk stage II and stage III colon cancer patients who received either biweekly or triweekly capeox. The rationale for the study is clear. The arms of the study are fairly well balanced and the results are presented appropriately. Limitations of the study are presented. Although T and N characteristics are presented it would be helpful to describe the numbers of Stage III A, B, C patients and high risk stage II patients and to describe which factors resulted in designating a stage II patient high risk. Although there were some differences in toxicities between the 2 regimens the differences are not striking and it is doubtful these results are practice changing.

Reviewer #3: The study titled "Biweekly CAPOX versus Triweekly CAPOX in the adjuvant therapy of post-surgery CRC: a randomized controlled trial" is a well-designed and clinically relevant investigation that addresses an important question in the management of colorectal cancer (CRC). The authors have conducted a randomized controlled trial to compare the safety and efficacy of a modified biweekly CAPOX regimen with the conventional triweekly CAPOX regimen in high-risk stage II and stage III post-surgery CRC patients. The study is particularly commendable for its focus on reducing treatment-related adverse events, such as thrombocytopenia and neutropenia, while maintaining therapeutic efficacy. The findings suggest that the biweekly CAPOX regimen is associated with fewer hematological adverse events and comparable 3-year disease-free survival (DFS) rates, which could have significant implications for improving patient tolerance and adherence to adjuvant chemotherapy. The study is well-written, methodologically sound, and provides valuable insights into optimizing adjuvant chemotherapy for CRC patients.

Comments for Revision:

1. While the study mentions that the sample size was calculated based on the primary outcome of thrombocytopenia, it would be beneficial to provide more details on the assumptions used for this calculation (e.g., expected incidence rates, effect size, power, and significance level). This will help readers understand the robustness of the study design and whether the sample size was adequate to detect clinically meaningful differences.

2. The study reports a median follow-up period of 42 months, which is commendable. However, it would be useful to discuss whether this follow-up duration is sufficient to capture long-term outcomes, particularly for DFS and overall survival (OS). Consider providing a rationale for the chosen follow-up period and whether longer follow-up is planned.

3. The study briefly mentions subgroup analyses based on treatment duration (3-month vs. 6-month). It would be valuable to provide more detailed results of these subgroup analyses, particularly in terms of adverse events and DFS. This could help identify specific patient populations that may benefit more from the biweekly regimen.

4. The study uses Kaplan-Meier estimation for DFS analysis and Cox proportional hazard models for hazard ratios (HRs). It would be helpful to clarify whether any adjustments were made for multiple comparisons, especially given the secondary endpoints. Additionally, consider discussing the potential impact of confounding variables on the results and whether any multivariate analyses were performed.

5. While the study reports a lower incidence of thrombocytopenia and neutropenia in the biweekly group, it would be beneficial to provide more detailed information on the severity and management of these adverse events. For example, how many patients required dose reductions, treatment delays, or supportive care interventions?

6. The discussion section could be expanded to include a more detailed discussion of the clinical implications of the findings. For instance, how might the reduced incidence of hematological adverse events impact patient quality of life, treatment adherence, and overall healthcare costs? Additionally, consider discussing how these findings might influence clinical guidelines or practice.

7. The study acknowledges that it is a pilot study with a limited number of patients. It would be helpful to elaborate on other potential limitations, such as the single-center design, potential selection bias, and the generalizability of the findings to other populations or settings.

8. The conclusion could be strengthened by suggesting directions for future research. For example, are there plans for a larger, multicenter trial to confirm these findings? Could future studies explore the use of biomarkers or molecular profiling to further personalize adjuvant chemotherapy regimens?

6. PLOS authors have the option to publish the peer review history of their article (what does this mean? ). If published, this will include your full peer review and any attached files.

**Do you want your identity to be public for this peer review?** For information about this choice, including consent withdrawal, please see our Privacy Policy .

Reviewer #1: No

Reviewer #2: No

Reviewer #3: **Yes: ** Mohammad Ebrahimnezhad

---

## [Author Response · Author response to Decision Letter 1]

11 Apr 2025

Response to Journal Requirements

Manuscript ID: PONE-D-24-47321

Title: Biweekly CAPOX versus Triweekly CAPOX in the adjuvant therapy of post-surgery CRC: a randomized controlled trial

We sincerely thank the editor for the time and effort dedicated to reviewing our manuscript. We truly appreciate your thoughtful comments and the opportunity to revise our work in accordance with the journal's requirements. We are also very pleased about the possibility of publishing our work in PLOS ONE.

1. Please ensure that your manuscript meets PLOS ONE's style requirements, including those for file naming. The PLOS ONE style templates can be found at https://journals.plos.org/plosone/s/file?id=wjVg/PLOSOne_formatting_sample_main_body.pdf and https://journals.plos.org/plosone/s/file?id=ba62/PLOSOne_formatting_sample_title_authors_affiliations.pdf.

Response:

Thank you for your comment. We have carefully reviewed and ensured that our manuscript meets all of PLOS ONE's style requirements, including those for file naming. We have referred to the provided templates and make several necessary adjustments to align with the journal's formatting guidelines.

2. Thank you for submitting your clinical trial to PLOS ONE and for providing the name of the registry and the registration number. The information in the registry entry suggests that your trial was registered after patient recruitment began. PLOS ONE strongly encourages authors to register all trials before recruiting the first participant in a study.

a) your reasons for your delay in registering this study (after enrolment of participants started);

b) confirmation that all related trials are registered by stating: “The authors confirm that all ongoing and related trials for this drug/intervention are registered.

Response:

Thank you for your helpful feedback. We acknowledge that the trial was registered after participant recruitment began. This delay was due to unforeseen logistical challenges, which led to the registration occurring after enrollment had already started. We have now included the following statement in the Methods section of our manuscript:

“The authors acknowledge that the trial was registered after participant recruitment started due to logistical delays. We confirm that all ongoing and related trials for this intervention are registered.”

This revision has been made in accordance with your editorial policy, and we appreciate your attention to this matter.

Response:

Thank you for your comment regarding the Data Availability Statement. We confirm that the submission contains all the raw data required to replicate the results of the study. As per PLOS ONE’s data availability policy, the minimal data set includes the values behind the means, standard deviations, and other reported measures, as well as the data used to build graphs and points extracted from images for analysis.

We have uploaded the relevant data as Supporting Information files. We have also included a detailed explanation in the Data Availability section regarding the minimal data set required to replicate the study’s findings.

Response:

Thank you for pointing this out. The first author and corresponding author now have ORCID iD and they have been validated in Editorial Manager. We have completed the process and can confirm that the ORCID iD is now included as required.

“This work was supported by Natural Science Foundation of Zhejiang Province, No. LQ23H160041.”

“Zhejiang Provincial National Science Foundation of China LQ23H160041 (to H.Z)”

Response:

We appreciate your comment about the placement of funding information in the Acknowledgments section. As per your guidance, we have removed all funding-related text from the Acknowledgments section. The correct funding information has been added to the Funding Statement section of the manuscript, as follows:

“This work was supported by the Zhejiang Provincial National Science Foundation of China, No. LQ23H160041 (to H.Z.).”

This updated statement has been reflected in the Cover Letter, as requested.

Response:

Thank you for your comment. We have revised the manuscript to ensure that the Ethics Statement only appears in the Methods section, as per the journal's requirements. Any instances of the ethics statement appearing outside of the Methods section have been removed.

Response to Reviewers

Manuscript ID: PONE-D-24-47321

Title: Biweekly CAPOX versus Triweekly CAPOX in the adjuvant therapy of post-surgery CRC: a randomized controlled trial

We thank all the Reviewers for their valuable comments and suggestions. We have carefully revised the manuscript accordingly. Below are our point-by-point responses.

Reviewer #1

Thank you for an interesting article that clearly outlines what was done and how to interpret the findings. I have a few comments to further clarify the article:

1. Comment 1:

Abstract: "This study aims" – please add an "s".

Response:

Thank you. We have corrected the phrase to "This study aims..." in the Abstract (line 25 of revised manuscript).

2. Comment 2:

Line 139: 0.44 effect size - isn't this a calculation based on a difference in proportions between the two groups? What proportions were used? was the proportion of all grade or grade 3+ the quantity of interest? Please provide a calculation that matches the incidence endpoint. Be clear if it is a rate estimated at any time during treatment, or if it is calculated from a time to event variable, which is possible if the AEs can take time to be experienced and there is differing times of follow up.

Response:

Thank you for pointing this out. We have clarified that the effect size was based on the difference in the incidence of all grade thrombocytopenia.

The sample size calculation was based on the primary outcome of all-grade thrombocytopenia. Drawing upon both previous literature1 and our center's clinical experience, we assumed an expected incidence of 60% for the triweekly CAPOX group. For the biweekly CAPOX group, based on preliminary pilot data and clinical observations, we estimated a reduction to 39%. The effect size (Cohen’s h = 0.44) was calculated using the arcsine transformation formula for two proportions. With a two-sided significance level of 0.05 and a power of 80%, we determined that 80 participants per group were required.

We confirm that adverse events were assessed from the start of treatment through 28 days after the last dose of CAPOX, consistent with standard AE monitoring windows in clinical trials. The primary outcome—incidence of all-grade thrombocytopenia—was calculated as a binary endpoint (event/no event) during this predefined observation period, not as a time-to-event variable. This definition has now been clarified in the Methods section (lines 130-135 of revised manuscript).

"Adverse events were monitored from the first day of treatment initiation until 28 days after the last dose of CAPOX. All events were assessed according to the Common Terminology Criteria for Adverse Events Version (CTCAE) version 4.0, and the incidence of all-grade thrombocytopenia during this observation window was recorded as the primary safety endpoint. "

Reference:

1 Jardim DL, Rodrigues CA, Novis YAS, Rocha VG, Hoff PM. Oxaliplatin-related thrombocytopenia. Ann Oncol. 2012;23(8):1937–1942. doi:10.1093/annonc/mds074.

3. Comment 3:

Non-inferiority is mentioned in the discussion - if the study was designed to test non-inferiority, please clearly indicate this in the sample size section of the paper.

Response:

Thank you for this important comment. We acknowledge that our study was not formally designed as a non-inferiority trial, and no non-inferiority margin was prespecified during the sample size calculation. To avoid confusion, we have removed the term “non-inferiority” from the Discussion section and have reframed our interpretation in terms of observed differences and clinical relevance. Please see lines 282–285 in the revised manuscript.

"However, as a pilot study with a limited number of patients, it lacks the statistical power to definitively demonstrate the superiority of biweekly CAPOX over triweekly CAPOX. Further studies are warranted to confirm the potential advantages of the biweekly regimen."

4. Comment 4:

What happened if a patient in the 3-month group experienced thrombocytopenia at 4 months - was that counted as an event? Be clear if the adverse events were collected within 6 months of starting treatment or just during study treatment.

Response:

Thank you for raising this point. Adverse events were systematically collected from the initiation of treatment until 28 days after the last dose of CAPOX, in accordance with standard clinical trial practice. Therefore, any thrombocytopenia occurring within 28 days after the final dose—including events in the 3-month group that happened during the 4th month—was included in the safety analysis. This time frame has been clarified in the Methods section of the revised manuscript (see lines 130–135 in the revised manuscript).

"Adverse events were monitored from the first day of treatment initiation until 28 days after the last dose of CAPOX. All events were assessed according to the Common Terminology Criteria for Adverse Events Version (CTCAE) version 4.0, and the incidence of all-grade thrombocytopenia during this observation window was recorded as the primary safety endpoint. "

5. Comment 5:

Figure 2F – state p > 0.99

Response:

We have updated Figure 2F and its legend to state 'p > 0.99'.

R1 Adverse events in biweekly and triweekly groups. (A) All grade thrombocytopenia in 3-month, 6-month and all patients between biweekly and triweekly group. (B) Grade more than 3 thrombocytopenia rate between biweekly and triweekly group. (C-D) All grade neutropenia and grade more than 3 neutropenia rate in 3-month, 6-month and all patients between biweekly and triweekly group. (E) Anemia rate in 3-month, 6-month and all patients between biweekly and triweekly group. (F) Peripheral sensory neuropathy rate in 3-month, 6-month and all patients between biweekly and triweekly group.

6. Comment 6:

The description of the results is valid.

Could 95% confidence intervals be added to the rates in Figures 2 and 3?

Response:

Thank you for your valuable comment. We have added the 95% confidence intervals (CIs) to the rates in Figure 2. Additionally, I have also provided the CI data for Figure 3 as shown in Figure R2 below for the detailed inclusion of the confidence intervals.

Furthermore, the description of the DFS data for both biweekly and triweekly groups, along with their respective 95% CIs, has also been included in the Results section, specifically on page 11, lines 197-212 in the revised manuscript.

" At the time of analysis, 29 events had been reported (13 in the biweekly group and 16 in the triweekly group). The Kaplan-Meier plots for DFS among biweekly group and triweekly group and subgroups are shown in Fig 3. The 3-year DFS rate was 85.1% [95% CI: 76.9%–93.3%] in biweekly group and 80.4% [95% CI: 71.2%–89.7%] in triweekly group (P = 0.51, HR = 0.78, [95%CI, 0.38-1.63]).

In the subgroup analysis of patients of 3-month treatment duration, the 3-year DFS was 89.2% [95% CI: 80.2%–98.2%] in biweekly group and 89.0% [95% CI: 79.8%–98.1%] in triweekly group. The difference was not statistically significant (P = 0.643). The HR for biweekly group to triweekly group was 0.89 [95% CI: 0.29–2.77], indicating no significant difference in DFS between the two groups.

For patients of 6-month treatment duration, the 3-year DFS was 77.6% [95% CI: 61.6%–93.6%] in biweekly group and 67.7% [95% CI: 50.3%–85.1%] in triweekly group, the HR was 0.79 [95% CI: 0.30–2.10] with no significant difference observed between the two groups ( P = 0.800). "

R2 Adverse events and disease-free survival (DFS) in biweekly and triweekly groups. (A) All grade thrombocytopenia in 3-month, 6-month and all patients between biweekly and triweekly group. (B) Grade more than 3 thrombocytopenia rate between biweekly and triweekly group. (C-D) All grade neutropenia and grade more than 3 neutropenia rate in 3-month, 6-month and all patients between biweekly and triweekly group. (E) Anemia rate in 3-month, 6-month and all patients between biweekly and triweekly group. (F) Peripheral sensory neuropathy rate in 3-month, 6-month and all patients between biweekly and triweekly group. (G) Overall disease-free survival (DFS) according to biweekly and triweekly CAPOX treatment groups.

Reviewer #2

The authors present their single center randomized phase II study for high risk stage II and stage III colon cancer patients who received either biweekly or triweekly capeox. The rationale for the study is clear. The arms of the study are fairly well balanced and the results are presented appropriately. Limitations of the study are presented.

7. Comment 1:

Although T and N characteristics are presented it would be helpful to describe the numbers of Stage III A, B, C patients and high risk st

---

## [Decision Letter · Decision Letter 1]

Biweekly CAPOX versus Triweekly CAPOX in the adjuvant therapy of post-surgery CRC: a randomized controlled trial

PONE-D-24-47321R1

Dear Dr. Zhang,

We’re pleased to inform you that your manuscript has been judged scientifically suitable for publication and will be formally accepted for publication once it meets all outstanding technical requirements.

Kind regards,

Keun-Yeong Jeong

Academic Editor

PLOS ONE

Reviewers' comments:

Reviewer's Responses to Questions

**Comments to the Author**

1. If the authors have adequately addressed your comments raised in a previous round of review and you feel that this manuscript is now acceptable for publication, you may indicate that here to bypass the “Comments to the Author” section, enter your conflict of interest statement in the “Confidential to Editor” section, and submit your "Accept" recommendation.

Reviewer #2: All comments have been addressed

2. Is the manuscript technically sound, and do the data support the conclusions?

Reviewer #2: Yes

3. Has the statistical analysis been performed appropriately and rigorously? 

Reviewer #2: Yes

4. Have the authors made all data underlying the findings in their manuscript fully available?

Reviewer #2: Yes

5. Is the manuscript presented in an intelligible fashion and written in standard English?

Reviewer #2: Yes

6. Review Comments to the Author

Reviewer #2: The authors have extensively addressed all the reviewers' comments and have made appropriate changes within the manuscipt

7. PLOS authors have the option to publish the peer review history of their article (what does this mean? ). If published, this will include your full peer review and any attached files.

**Do you want your identity to be public for this peer review?** For information about this choice, including consent withdrawal, please see our Privacy Policy .

Reviewer #2: No

---

## [Editor Report · Acceptance letter]

PONE-D-24-47321R1

PLOS ONE

Dear Dr. Zhang,

I'm pleased to inform you that your manuscript has been deemed suitable for publication in PLOS ONE. Congratulations! Your manuscript is now being handed over to our production team.

Kind regards,

on behalf of

Dr. Keun-Yeong Jeong

Academic Editor

PLOS ONE